# RFRP-3 Influences Apoptosis and Steroidogenesis of Yak Cumulus Cells and Compromises Oocyte Meiotic Maturation and Subsequent Developmental Competence

**DOI:** 10.3390/ijms24087000

**Published:** 2023-04-10

**Authors:** Xianrong Xiong, Yulei Hu, Bangting Pan, Yanjin Zhu, Xixi Fei, Qinhui Yang, Yumian Xie, Yan Xiong, Daoliang Lan, Wei Fu, Jian Li

**Affiliations:** 1Key Laboratory of Qinghai-Tibetan Plateau Animal Genetic Resource Reservation and Exploitation of Ministry of Education, College of Animal & Veterinary Sciences, Southwest Minzu University, Chengdu 610041, China; xianrongxiong@swun.edu.cn (X.X.);; 2Key Laboratory for Animal Science of National Ethnic Affairs Commission, College of Animal & Veterinary Sciences, Southwest Minzu University, Chengdu 610041, China

**Keywords:** RFRP-3, apoptosis, steroidogenesis, cumulus cell, development

## Abstract

RF amide-related peptide 3 (RFRP-3), a mammalian ortholog of gonadotropin-inhibitory hormone (GnIH), is identified to be a novel inhibitory endogenous neurohormonal peptide that regulates mammalian reproduction by binding with specific G protein-coupled receptors (GPRs) in various species. Herein, our objectives were to explore the biological functions of exogenous RFRP-3 on the apoptosis and steroidogenesis of yak cumulus cells (CCs) and the developmental potential of yak oocytes. The spatiotemporal expression pattern and localization of GnIH/RFRP-3 and its receptor GPR147 were determined in follicles and CCs. The effects of RFRP-3 on the proliferation and apoptosis of yak CCs were initially estimated by EdU assay and TUNEL staining. We confirmed that high-dose (10^−6^ mol/L) RFRP-3 suppressed viability and increased the apoptotic rates, implying that RFRP-3 could repress proliferation and induce apoptosis. Subsequently, the concentrations of E_2_ and P_4_ were significantly lower with 10^−6^ mol/L RFRP-3 treatment than that of the control counterparts, which indicated that the steroidogenesis of CCs was impaired after RFRP-3 treatment. Compared with the control group, 10^−6^ mol/L RFRP-3 treatment decreased the maturation of yak oocytes efficiently and subsequent developmental potential. We sought to explore the potential mechanism of RFRP-3-induced apoptosis and steroidogenesis, so we observed the levels of apoptotic regulatory factors and hormone synthesis-related factors in yak CCs after RFRP-3 treatment. Our results indicated that RFRP-3 dose-dependently elevated the expression of apoptosis markers (*Caspase* and *Bax*), whereas the expression levels of steroidogenesis-related factors (*LHR*, *StAR*, 3*β*-HSD) were downregulated in a dose-dependent manner. However, all these effects were moderated by cotreatment with inhibitory RF9 of GPR147. These results demonstrated that RFRP-3 adjusted the expression of apoptotic and steroidogenic regulatory factors to induce apoptosis of CCs, probably through binding with its receptor GPR147, as well as compromised oocyte maturation and developmental potential. This research revealed the expression profiles of GnIH/RFRP-3 and GPR147 in yak CCs and supported a conserved inhibitory action on oocyte developmental competence.

## 1. Introduction

The development of follicles is essentially controlled by reproductive hormones from the hypothalamus–pituitary–ovary axis, impairing reproductive efficiency. The follicle in the mammalian ovary, which contains an oocyte and surrounded soma cells, is the fundamental functional unit for natural oocyte growth and steroidogenesis [1]. Hormone secretion is the primary physiological function of these soma cells around oocytes. Cumulus cells (CCs) are one of these soma cells in follicles and can provide numerous small molecular substances which are necessary for oocyte growth and maturation [1,2]. The proliferation and apoptosis of CCs influence the production of steroid hormones in vivo [3]. The ability of steroidogenesis in CCs, to a certain extent, determines the fate of follicular development to ovulation or atresia. Mimicking the in vivo microenvironment, in vitro maturation (IVM) of immature oocytes is considered an effective method to obtain matured oocytes for practical applications, especially in human-assisted reproduction. At present, the developmental potential and fertilization ability of in vitro matured oocytes are significantly low compared with those of naturally matured oocytes in vivo [4]. This is because the oocytes derived in vivo have adequate crosstalk with surrounding soma cells, such as CCs, and the communication channels between oocytes and soma cells are destroyed when the germinal vesicle (GV) oocyte is in IVM. Therefore, exploring the communication between CCs and oocyte is essential for improving reproductive efficiency and solving infertility in mammals [5]. Research on the interactions between CCs and oocytes may help improve understanding of the underlying mechanism of follicular development and eventually improve the efficiency of in vitro embryo production (IVP).

The processes of follicular development and oocyte maturation in vivo are rather complicated and are adjusted by many factors, such as hypothalamic neuropeptides [6]. The gonadotropin-releasing hormone (GnRH), a common neuropeptide with pronounced promotional actions, can effectively regulate the growth of oocytes and follicles by stimulating the pituitary gland to synthesize and release follicle-stimulating hormone (FSH) and luteinizing hormone (LH) [6]. Remarkable advances have been made in elucidating that GnRH plays an intrinsic effect in the hypothalamus–pituitary–gonadal (HPG) axis, but it is not the unique neuropeptide in charge of female breeding [7]. As a new type of neuropeptide with inhibitory actions, gonadotropin-inhibiting hormone (GnIH) was first found in Japanese quail, and it inhibits the synthesis and release of gonadotropic peptides [8]. GnIH is a key neuropeptide that involves negatively regulating reproductive functions in various species, such as *zebrafish* (lower vertebrates), murine (higher vertebrates), and primates [9,10]. In addition, GnIH plays many biological functions besides regulating animal reproduction, including food intake and energy homeostasis [11,12]. Recent studies have further indicated that GnIH affected the hypothalamus and pituitary to regulate sexual reproduction, as well as the behavior of reproduction via altering neurohormone biosynthesis in the brain. The abnormal expression of GnIH may lead to reproductive dysfunction or puberty disorder [13,14]. Piekarski et al. [15] also showed that GnIH is a crucial regulatory factor of sexual behavior and motivation in female hamsters. These mammalian findings indicated that GnIH does not only regulate the reproductive axis but also controls socially motivated behavior.

On the basis of a common C-terminal LPXRFamide structure, GnIHs are categorized as LPXRFamide peptides and named RFRP-1 and -3 (RFamide-related peptides 1 and 3) in mammals. RFRP-3, as a structure and functional ortholog of GnIH, is identified to be a new inhibitory neurohormonal peptide of breeding, which plays a crucial function through the reproductive axis across a broad range of species [13,16]. RFRP-3 mainly occurs in the hypothalamic neurons, pituitary glands, and gonads of vertebrates [17]. RFRP-3 was found to inhibit the release and synthesis of LH and gonadotropin in rats [18], sheep [19], and bovines [20]. Interestingly, in primates, RFRP-3 suppresses the release of FSH and LH and reduces LH and FSH subunit expression; thus, RFRP-3 has a similar function in inhibiting the synthesis and release of GnRH-stimulated gonadotropin in mammals [13,16]. Furthermore, RFRP-3 was identified in ovaries across nearly all vertebrates, from lampreys to humans [16,21], but studies on the specific function of RFRP-3 in ovaries are few. RFRP-3 is primarily mediated with its receptor (GPR147, also known as NPFF1) to exert biological functions, such as controlling gonadotropin release and synthesis [13,16]. In mouse LβT2 cells and GT1-7 cell line, GnIH/RFRP-3 suppresses the secretion of GnRH-induced gonadotropin through the ERK pathway, which relies on AC/cAMP/PKA and vasoactive intestinal polypeptide [22]. Moreover, a previous study showed that RFRP-3 directly influences the gonads to reduce the synthesis of sex steroids and the differentiation and maturation of germ cells [23]. To date, RFRP-3 decreases cell viability and steroidogenesis in the ovaries of chickens, pigs, mice, and humans [23,24,25,26]. Other work also suggested that the relative expression abundance of RFRP-3 in the gonads is related to reproductive performance [27]. In addition, the location and expression profiles of RFRP-3 and its receptors imply a crucial role in steroidogenesis and gametogenesis [26,28]. The occurrence of RFRP-3 and its receptors in numerous species suggested that RFRP-3 plays a crucial regulatory role in the control of ovarian activities [24,29], such as ovulation, follicular development, and steroid secretion. A recent study on mice indicated that ovarian activity and follicular growth are suppressed after treatment with RFRP-3 [26]. Immunohistochemistry (IHC) found that RFRP-3 and its receptors are primarily expressed in GCs and membrane cells of bird and pig ovarian follicles [25,30], and GnIH/RFRP-3 plays a crucial regulatory effect in the ovaries via combining with its receptors, such as adjusting apoptosis, proliferation, and steroidogenesis [31]. However, as the main domestic animal in the Qinghai-Tibet Plateau region and nearby areas, whether the low reproductive efficiency of female yak is related to GnIH/RFRP-3 remains unclear and needs further investigation, as well as to evaluate whether RFRP-3 exerts conserved function across various species.

Thus, we hypothesized that GnIH/RFRP-3 could be closely associated with the biological activities of yak follicular cells and oocytes, resulting in follicular developmental suppression and low reproductive efficiency. In this study, to investigate the biological functions of RFRP-3 on yak CCs and oocytes, we utilized IHC to define whether GnIH/RFRP-3 and its receptor GPR147 were ubiquitously expressed in yak ovary and separate soma cells. The effects of various dosages of exogenous RFRP-3 on apoptosis and steroidogenesis in yak CCs were evaluated via TUNEL staining and enzyme-linked immunosorbent assay (ELISA), respectively. In addition, the nuclear maturation dynamics of yak oocytes during IVM and subsequent developmental potential were also confirmed after RFRP-3 treatment. Finally, to lay out the corresponding action mechanisms, the effects of RFRP-3 on the expression of genes regulating apoptosis and steroidogenesis in yak CCs were assessed. These findings provided favorable evidence for the hypothesis that endogenous GnIH/RFRP-3 is an important component for regulating the development of follicles and oocytes in yak. Simultaneously, our study could be beneficial to understand the reproductive function of RFRP-3 and helpful for establishing new strategies used for the diagnosis and treatment of reproductive diseases, animal breeding control, and endangered species survival programs.

## 2. Results

### 2.1. Expression Profiles of GnIH/RFRP-3 and GPR147 in Yak Ovary and CCs

The expression patterns and subcellular localization of GnIH/RFRP-3 and GPR147 were investigated during yak folliculogenesis. The IHC results showed that GnIH/RFRP-3 and GPR147 were abundantly distributed in oocytes and around granulosa cells and theca cells of various stages of follicles, such as primordial follicles, primary follicles, preantral follicles, antral follicles, and corpus luteum (shown in Figure 1). These data suggested a potential function of GnIH/RFRP-3 and GPR147 during oocyte growth and follicular development, although the specific mechanism by which GnIH/RFRP-3 and GPR147 regulate the growth of follicles remains unclear. In addition, the relative abundance of GnIH/RFRP-3 and GPR147 was dynamically changed during follicular development.

To study the subcellular specific distribution of GnIH/RFRP-3 and GPR147 in primary CCs, the specific antibodies of GnIH/RFRP-3 and GPR147 were used for immunofluorescence. FSHR, a specifically expressed protein in ovarian CCs, was used to identify the purity of cultured cells. The results showed that almost all cells could bind to the FSHR antibody (Figure 2), suggesting that yak ovarian CCs were successfully isolated and cultured. Interestingly, GnIH/RFRP-3 was specifically expressed in the cytoplasm, but low expression in the nucleus was found in CCs (Figure 2). As GnIH/RFRP-3 was primarily bound to GPR147, immunofluorescence was used to confirm whether GPR147 is expressed in yak CCs. Our results indicated that GPR147 was evenly expressed and mainly localized in the cytoplasm of CCs.

### 2.2. RFRP-3 Inhibits the Proliferation of Yak CCs

To confirm the effect of RFRP-3 on cell viability, yak CCs were treated with different doses (0, 10^−10^, 10^−8^, and 10^−6^ mol/L) of RFRP-3 at different times (12, 24, and 36 h), and then viability was evaluated with CCK-8 assay. The result showed that the cell viability of yak CCs was inhibited with RFRP-3 in a dose-dependent manner, and 10^−6^ mol/L RFRP-3 significantly decreased cell viability (*p* < 0.05, Figure 3A). After 24 h of treatment with RFRP-3, the absorbance was the highest compared with the other groups; thus, the CCs were incubated with 10^−6^ mol/L RFRP-3 for 24 h in subsequent experiments.

To investigate the effects of RFRP-3 and GPR147 on the proliferation of yak CCs, CCs were treated with 10^−6^ mol/L RFRP-3 and/or 10 μM RF9 for 24 h. The results showed that the proliferation competence of yak CCs decreased significantly in the 10^−6^ mol/L RFRP-3 group (*p* < 0.01) compared with the controls (Figure 3B,C). However, combined treatment with 10 μM RF9 effectively defused the inhibitory role of RFRP-3 and showed no significant difference compared with the control counterparts (*p* > 0.05). Moreover, the mRNA expression levels of three marker genes (*Cdc42*, *Ccnd1*, and *Pcna*) of cell proliferation were evaluated by RT-qPCR. RFRP-3 dramatically reduced the relative expression abundances of *Cdc42*, *Ccnd1*, and *Pcna* (Figure 3D–F), suggesting that 10^−6^ mol/L RFRP-3 inhibited CC proliferation. RF9 supplementation upregulated the mRNA abundances of *Cdc42* and *Ccnd1* (*p <* 0.05) but not *PCNA*. The above data suggested that the proliferation activity of yak CCs was associated with RFRP-3.

### 2.3. RFRP-3 Promotes Apoptosis with Apoptosis-Related Genes

After 24 h incubation with various dosages of RFRP-3, the apoptotic rates of yak CCs increased in a dose-dependent manner (Figure 4A). Treatment with 10^−6^ mol/L RFRP-3 notably increased the apoptosis of yak CCs (*p <* 0.05). However, there was no remarkable difference in the low-dose (10^−10^ mol/L) RFRP-3-treated group compared with the controls (*p >* 0.05). Furthermore, the apoptotic rate predominantly decreased (*p* < 0.05) in the CCs co-treated with 10^−6^ mol/L RFRP-3 and 10 μM RF9 than that in the only CCs subjected to 10^−6^ mol/L RFRP-3 treatment, whereas no significant difference (*p* > 0.05) with the other treatment groups was noted. To explore the underlying mechanism of the effects of RFRP-3 on yak CC apoptosis, RT-qPCR was employed to analyze the expression of apoptosis-related genes, including *Caspase-3*, *Bax*, and *Bcl-2*. These results suggested that the treatment of high (10^−6^ mol/L) and moderate (10^−8^ mol/L) doses of RFRP-3 significantly upregulated the relative expression level of *Caspase-3* (*p <* 0.05), but no remarkable difference was found between the 10^−10^ mol/L group and its control counterpart (*p >* 0.05, Figure 4B,C). Coincidentally, the mRNA expression patterns of *Bax* changed similarly to those of *Caspase-3*. Specifically, *Bax* mRNA was upregulated in a dose-dependent manner with RFRP-3 treatment (Figure 4D). On the contrary, the relative expression abundance of *Bcl-2* presented an opposite pattern from that of *Caspase-3* mRNA in the RFRP-3-supplied group. Treatment with high (10^−6^ mol/L) and moderate (10^−8^ mol/L) doses of RFRP-3 notably downregulated the expression of *Bcl-2*, whereas treatment with 10^−10^ mol/L RFRP-3 resulted in no remarkable difference compared with the control counterparts (*p >* 0.05). Nevertheless, RF9 significantly suppressed the high-dose (10^−6^ mol/L) RFRP-3-induced increase of *Caspase-3* and *Bax* expression, whereas the relative mRNA abundance of *Bcl-2* was upregulated (*p* < 0.05) after cotreatment with 10^−6^ mol/L RFRP-3 and RF9.

### 2.4. Effects of RFRP-3 on E_2_ and P_4_ Secretion

The spatial and temporal expression of GnIH and its receptor GPR147 in the yak ovary revealed that the GnIH/RFRP-3/GPR147 axis participated in the biological activities of yak ovarian tissue. The isolated yak CCs were used to evaluate the influence of RFRP-3 on the secretion of P_4_ (progesterone) and E_2_ (17β-estradiol) by ELISA assay (Figure 5A,B). The contents of P_4_ and E_2_ in the supernatant of CCs treated with low-dose (10^−10^ mol/L) RFRP-3 were not significantly different from those of their control counterparts (*p* > 0.05). Incubation with moderate (10^−8^ mol/L) and high doses (10^−6^ mol/L) of RFRP-3 resulted in a considerable reduction in E_2_ content (*p* < 0.05), but high-dose (10^−6^ mol/L) of RFRP-3 led to a significant downregulation in P_4_ concentration (*p* < 0.05). Interestingly, RF9 combined with RFRP-3 (10^−6^ mol/L) notably upregulated the concentrations of E_2_ and P_4_ compared with RFRP-3 (10^−6^ mol/L) treatment only (*p* < 0.05).

As three markers associated with predominant regulation of steroidogenesis (*LHR*, *StAR*, 3*β*-HSD), their mRNA relative expression levels were detected by RT-qPCR after RFRP-3 and/or RF9 treatment. The results reflected a remarkable change in *LHR* (Figure 5C) and *StAR* (Figure 5D) but a finite alteration in *3β-HSD* (Figure 5E) in yak CCs after various dosages of RFRP-3 treatment. Specifically, the mRNA relative abundance of *LHR* exhibited a notable dose-dependent reduction (*p* < 0.05 or *p* < 0.01) compared with the control group. The groups treated with 10^−10^ mol/L and 10^−8^ mol/L RFRP-3 demonstrated a mild (*p* < 0.05) decrease, but the 10^−6^ mol/L group presented a conspicuous downregulation (*p* < 0.01) in *StAR* mRNA abundance compared with the control counterparts. The 10^−10^ mol/L and 10^−8^ mol/L RFRP-3 groups showed no dramatic variation (*p* > 0.05), whereas high-dose (10^−6^ mol/L) of RFRP-3 significantly reduced the expression of *3β-HSD* compared with the controls (*p* < 0.05). Moreover, the relative expression levels of *LHR* and *StAR* were predominantly elevated (*p* < 0.05), but no variation in *3β-HSD* was found after cotreatment with RFRP-3 (10^−6^ mol/L) and RF9 (*p* > 0.05) compared with that of the RFRP-3 (10^−6^ mol/L) only group. Together, the above data indicated that RFRP-3 suppressed E_2_ and P_4_ secretion in yak CCs by inhibiting the expression of factors related to hormone synthesis.

### 2.5. Effects of RFRP-3 on the Progression of Oocyte Maturation

First, the localization of GnIH/RFRP-3 and its receptor GPR147 in oocytes was determined. Both GnIH/RFRP-3 and GPR147 were significantly transcribed in the oocyte cytoplasm (Figure 6A). To explore the influence of RFRP-3 on the meiotic resumption of yak oocytes, we evaluated mature oocytes at four stages: GV (germinal vesicle), GVBD (germinal vesicle breakdown), MI (metaphase I), and MII (metaphase II). After 6 h of IVM, the group supplemented with 10^−6^ mol/L RFRP-3 showed a significant decrease in the GVBD rate compared with the controls (*p* < 0.05, Figure 6B–D), but no remarkable difference was observed in oocytes following treatment with 10^−10^ mol/L RFRP-3 (*p* > 0.05). Nevertheless, supplementation with RF9 dramatically enhanced the GVBD rate compared with supplementation with 10^−6^ mol/L RFRP-3 (*p* < 0.05). Similar to GVBD results, treatment with 10^−6^ mol/L RFRP-3 individually decreased MI response, and cotreatment with RF9 paradoxically reduced the RFRP-3-induced MI response compared with 10^−6^ mol/L RFRP-3 treatment alone (Figure 6D). After 24 h of IVM, the group supplemented with 10^−6^ mol/L RFRP-3 exhibited a markedly (*p* < 0.05) lower MII rate than the controls. By contrast, no considerable difference was found between the other RFRP-3-treated groups and the control group (*p* > 0.05).

### 2.6. Effects of RFRP-3 on the Oocyte Developmental Potential

To further verify the effects of RFRP-3 on oocyte developmental competence, mature oocytes from different treatments were parthenogenetically activated, and subsequent development at three stages, namely, 2-cell (cleavage, 24 h), 8-cell (72 h), and blastocyst (168 h), was evaluated. The 2-cell, 8-cell, and blastocyst formation rates were severely lower in the group of 10^−6^ mol/L RFRP-3-treated oocytes than in their control counterparts (*p* < 0.05, Figure 6E). By contrast, the blastocyst formation rate was significantly higher in the low-dose (10^−10^ mol/L) RFRP-3-treated oocyte group than in the other groups (*p* < 0.05). However, supplementation with RF9 significantly reversed the inhibitory effects of RFRP-3 (10^−6^ mol/L) on the developmental potential of oocytes to form blastocysts. Therefore, RFRP-3 exerted bidirectional effects on the developmental competence of yak oocytes.

## 3. Discussion

GnIH, a t12-amino acid residue neuropeptide hormone, is concerned with numerous behavioral and physiological functions in animal reproduction [10], including the suppression of steroidogenesis and follicular growth in female ovarian tissues [26,30,31]. The reproductive performance of mammals is primarily affected by the number of mature follicles that can develop to ovulation. However, to ensure the competence for fertilization and subsequent embryo development of the ovulated oocytes in mammals, most follicles in female ovarian tissues begin to exhibit atresia before and after birth [32]. The apoptosis of soma cells in follicles, especially surrounding CCs, is closely associated with follicular atresia in vivo [33]. In this study, we first reported the physiological function and corresponding molecular underlying mechanism of RFRP-3 on yak CCs and oocytes. Our results suggest that exogenous RFRP-3 impairs the proliferation, apoptosis, and steroidogenesis of yak CCs, which may influence the progression of oocyte maturation. The resulting oocytes may fail to fully mature, thereby affecting the subsequent developmental potential.

The accumulating evidence demonstrated that RFRP-3 was abundantly expressed in both the brain and the gonads and played pivotal roles in suppressing animal reproductive ability [13,16]. This study revealed that GnIH/RFRP-3 and GPR147 were ubiquitously expressed in yak ovarian tissues, primarily in the theca cells and granulosa cells of preantral and antral follicles. Previous studies indicated that GnIH/RFRP-3 and its receptors are expressed in germ cells and steroidogenic cells in the testicular and ovarian tissues of birds and mammals [23,27,28]. Although the expression of RFRP was almost undetectable in the chicken ovaries [24], a significant difference was noted in the relative expression abundances of RFRP and GPR147 in soma cells of different follicular stages in mice and pigs [25,26]. All these reports revealed that spermatogenesis or follicular development is regulated in the gonadal tissues, although the specific mechanisms need to be further explored. The immunolocalization of GnIH/RFRP-3 and its receptor GPR147 indicated that they were dynamically expressed in pig ovarian cells during the estrous cycle [25]. Consistent with previous studies on mice and Calotes versicolor, our results further demonstrated that GnIH/RFRP-3 and GPR147 were primarily expressed in the follicular theca cells and granulosa cells of matured follicles, as well as the luteal cells of yak ovaries [34,35]. Meanwhile, we speculated that the relative mRNA expression levels of GnIH/RFRP-3 and GPR147 in the ovary were negatively correlated with the developmental fate of follicles. In other words, GnIH/RFRP-3 and GPR147 may be responsible for the fate of a follicle to grow or develop atresia. Additionally, a dramatic upregulation of GnIH/RFRP-3 during diestrus is likely involved in follicular development and selection, as well as corpus luteum activity [31]. The present study also revealed that GnIH/RFRP-3 and GPR147 were dynamically expressed in the various stages of follicles and corpus luteum in yak ovaries. Consistent with this finding, GnIH/RFRP-3 and GPR147 were spatially and temporally expressed in other vertebrates’ ovaries [24,29]. Moreover, the dynamic expression of *RFRP-3* was remarkably different in various species, with slightly high expression in the hypothalamus and pituitary of humans but abundantly expressed in pigs [23,36]. Our study confirmed that *GnIH*/*RFRP-3* and its receptor *GPR147* were mainly expressed in yak ovarian cells, and they might regulate the ovarian steroidogenesis function in an autocrine and/or paracrine manner. Nevertheless, the specific physiological significance and biological function of GnIH/RFRP-3 on yak follicular development remains unclear and needs further study.

GnIH/RFRP-3 plays crucial roles in multiple parts of the reproductive axis in mammals. Recent studies revealed that exogenous RFRP-3 plays a part in the female pig HPG axis in vitro and inhibits the proliferation of porcine ovarian cells by blocking these cells at the G2/M stage [25,37]. The present study indicated that RFRP-3 repressed the viability and proliferation of yak CCs and decreased the mRNA abundance of proliferation-related factors. In agreement with our findings, RFRP-3 also conspicuously downregulated the transcription of proliferation-related factors in porcine GCs in vitro [38]. RFRP-3 drastically reduced the expression of CDC42, PCNA, and CCND1 in mRNA levels to inhibit bovine GC proliferation [31]. Therefore, RFRP-3 has a negative effect on the proliferation of yak CCs, although we cannot rule out the possibility of an underlying mechanism.

Apoptosis of ovarian cells, to a certain extent, is closely related to the atresia of follicles [39], and it is adjusted by a complex regulatory network [40]. Therefore, studies on the apoptosis and corresponding action mechanism of CCs may be beneficial to elaborate the follicular development and reproductive characteristics of yak. Apoptosis is triggered by numerous factors, including apoptosis-related genes and hormones [41]. Apoptosis is a form of cell death characterized by a series of specific biochemical and morphological alterations, which are mainly caused by apoptosis-related factors [42]. Recent reports indicated that GnIH/RFRP-3 mediates autophagy and apoptosis of epididymal cells in rats and suppresses GC proliferation in pigs. Similarly, in cattle and porcine, GnIH/RFRP-3 inhibits GC proliferation, damages mitochondrial function, and promotes GC apoptosis by notably decreasing the p38 phosphorylation level [31,37,38]. In male mammals, GnIH/RFRP-3 induces testicular and epididymal apoptosis via mediating p53, Bax, and Capase-3 [43]. The present study was the first to report that RFRP-3 mediated the apoptosis of yak CCs in vitro. Specifically, 10^−6^ mol/L RFRP-3 caused a dramatic upregulation in the mRNA abundance of pro-apoptotic genes but a considerable reduction in the mRNA relative expression of the antiapoptotic marker. The above results indicated that RFRP-3 induced yak CC apoptosis in a dose-dependent manner. By contrast, Zhang et al. [38] found a bidirectional manner of RFRP-3 on apoptotic factors, as treatment with high-dose (10^−6^ M) RFRP-3 induced the upregulation of apoptotic factors, and treatment with low-dose (10^−12^ M) RFRP-3 induced an inverse trend. Possible explanations for the discrepancy could be the differences in species, experimental conditions, and times of treatment. Thus, RFRP-3 is involved in the apoptosis of yak CCs by upregulating pro-apoptotic members and suppressing anti-apoptotic factors.

Even though the expression profiles of GnIH/RFRP-3 and GPR147 have been described in ovaries of numerous kinds of species, their specific biological functions in the steroidogenesis of the ovaries are limited. The results of this study indicated that RFRP-3 played a direct role in the steroidogenesis of yak CCs. Consistent with our study, GnIH/RFRP-3 treatment decreased the accumulation of P_4_ and the expression of *3β-HSD* and *StAR* [26]. In isolated human granulosa cells, RFRP-3 treatment reduced P_4_ accumulation and the expression of *StAR* [23]. Unlike in mouse ovaries, RFRP-3 treatment has no significant effect on the expression of *3β-HSD* in human ovarian cells [23]. These differential effects of RFRP-3 on steroidogenesis could be caused by the different stages of folliculogenesis and various species [36,44]. Previous studies also showed that a low dose (100 ng per day) of GnIH/RFRP-3 treatment induces no remarkable variation in E_2_ and P_4_ levels in mice, whereas high-dose (2 μg per day) GnIH/RFRP-3 causes a notable increase in P_4_ and E_2_ concentration [23,26]. The improved concentration of circulating E_2_ could stimulate fat accumulation in the GnIH/RFRP-3-induced mouse [13]. Data from rats revealed that exogenous GnIH/RFRP-3 did not affect the secretion of E_2_ and P_4_, but it inhibited the gonadotropin-induced secretion of these two hormones [45]. These contradictory results in P_4_ and E_2_ contents could be due to the use of different species and/or experimental conditions. Although growing evidence implicated that RFRP-3 regulates the secretion of gonadal hormones, the specific steroidogenic pathway of RFRP-3 has yet to be elucidated. Alternatively, RFRP-3 plays its distinctive role by regulating the expression of steroidogenesis-related genes.

Therefore, we investigated the crucial genes related to steroidogenesis and showed that these key factors were severely changed in a dose-dependent manner upon RFRP-3 supplementation. RFRP-3 significantly downregulated *LHR* and *StAR* levels, and the *3β-HSD* level was significantly reduced only in the 10^−6^ mol/L RFRP-3 group. Interestingly, a previous study indicated that GnIH/RFRP-3 suppresses follicular development by decreasing the secretion of reproductive hormones [46]. Thus, we speculated that RFRP-3 inhibited follicular growth by increasing the GnIH concentration in vivo and blocked the crucial pathway of steroidogenesis mostly via reducing the mRNA abundances of *LHR*, *3β-HSD*, and *StAR*. Similarly, granulosa cells of quail and duck showed a considerable dose-dependent reduction of the expression levels of the core factors in the steroidogenic pathway [30,47]. Consistent with our findings, treatment with GnIH dose-dependently decreased the mRNA abundances of *StAR* and *3β-HSD* and repressed the synthesis of the steroid hormone in mice [26]. The defects of the present study were that we only conducted experiments in vitro, so the roles of RFRP-3 on steroidogenesis and follicular development in vivo still need to be further investigated.

The accumulating findings revealed that follicular development and oocyte maturation are regulated by estradiol and the gonadal peptides, such as GnRH and GnIH, as well as exogenous factors [48]. In general, oocytes are blocked in the G2 stage of MI until the GnRH-induced resumption of meiosis, which includes a switch from prophase-I to MII, while the ovulated oocyte is arrested again until fertilization [49]. During this complex process, GnRH and GnIH play a pivotal role in the regulation of the meiotic resumption in vivo [48]. This study was conducted to explore the direct action of RFRP-3 on the progression of oocyte meiotic maturation in vitro. With regard to GnIH/RFRP-3, our data indicated that RFRP-3 inhibited GVBD and meiotic maturation of yak oocytes in a dose-dependent manner, as well as bidirectional roles on the subsequent developmental competence. Similarly, treatment with GnIH stimulated the GVBD of the zebrafish oocytes, and the activity of caspase-3 was notably changed in early- and mid-stage oocytes [48]. In mice, Singh et al. [26] also demonstrated that treatment with GnIH impairs ovarian follicular morphology and development in a dose-dependent manner. Along with the increase in the GnIH concentration, the high-quality follicles that possessed potential ovulation ability markedly decreased, and the number of atretic follicles significantly increased, which indicated that GnIH effectively suppressed the development of follicles in vivo [26]. To date, the content of RFRP-3 in ovaries has never been detected in any species, and further work will be needed to investigate the biological contents of GnIH/RFRPs during the follicular development in yak ovaries.

At present, little information is available on the intracellular signal pathways of GnIH/RFRPs in multifarious ovarian activities. Previous studies confirmed that the GnIH receptor GPR147, also named NPFF receptor 1, is considered a novel member of the G-protein-coupled receptors [50]. The interaction between GnIH/RFRP-3 and GPR147 is known from studies of their mRNA and protein expression profiles in cat and rat ovarian cells [45,51]. Biochemical examination demonstrated that RFRP-3 has a greater affinity with GPR147, so GPR147 is one of the major receptors for RFRP-3 [50,52]. In this study, we used GPR147 antagonist RF9 to evaluate the intracellular pathway of RFRP-3. We found that the progression of oocyte meiotic maturation was resumed by the supplementation of RF9, suggesting that RFRP-3, to a certain degree, interacted with GPR147 to affect oocyte maturation and developmental competence. However, we inferred that the rhythm of cumulus–oocyte communication was attenuated during IVM and even had no significant change on GVBD, which could be harmful to subsequent developmental potential. Although our study proved that GnIH/RFRP-3 interacted with the GPR147 receptor, further exploration of the various functions of RFRP-3 in different species is beneficial to establish novel diagnostic and therapeutic programs for infertility and subfertility.

## 4. Conclusions

Our study implicated that exogenous RFRP-3 has a noteworthy role in the cumulus cell–oocytes in vitro. As a multifunctional regulatory factor, RFRP-3 impairs the proliferation and apoptosis of yak CCs, suppresses the synthesis and release of E_2_ and P_4_, and shows an inhibitory effect on oocyte maturation and developmental potential. Our study demonstrated that RFRP-3 plays a vital role in the yak reproductive axis in vitro and effectively regulates the proliferation, apoptosis, and steroidogenesis of yak CCs, as well as oocyte maturation and subsequent development. All these facts strongly demonstrated that GnIH/RFRP-3 in follicles is an important component of the complex multifactorial regulatory system, which regulates yak CC growth and oocyte maturation in vitro. Thus, the exact relevance and action network of CC apoptosis and oocyte maturation under exogenous RFRP-3 in various species and conditions are worthy of further investigation.

## 5. Materials and Methods

### 5.1. Animals and RFRP-3

All experiments in this study were performed in strict accordance with the guidelines of the Southwest Minzu University Animal Ethics and Use Committee and approved by the Southwest Minzu University Ethical Committee (approval number: 2020A017). The samples were collected in Hongyuan County, Sichuan Province, China (longitudinal 32°78′ N latitudinal 102°52′ E). A total of 24 female yaks (3-year-old, non-pregnant, and never pregnant delivery before) were selected in this study. All animals have similar growth environments and equivalent nutritional conditions.

A GRP147 antagonist, RF9, was purchased from Tocris Bioscience Co. (RF9, catalog No. 3672, Ellisville, MO, USA). Human RFRP-3 was purchased from Phoenix Pharmaceuticals Co. (Val-Pro-Asn-Leu-Pro-Gln-Arg-Phe-NH2, Catalog No. 048-46, Burlingame, CA, USA). Aliquots of RFRP-3 peptide were dissolved with ultrapure water and stored at −20 °C for further use.

### 5.2. Immunohistochemistry (IHC) of Ovarian Tissue

Fixed ovaries were paraffin-embedded, sectioned, and subjected to IHC, as described in a previous study [53]. In brief, all 5 µm sections were deparaffinized and rehydrated in a graded series of ethanol (100–70%). Subsequently, 10 mM sodium citrate (pH = 6.0) was used for antigen retrieval, and 0.2% TritonX-100 was used for permeabilization. The sections were respectively incubated with primary antibodies GnIH (N2176-47, 1:500, Biological, Swampscott, MA, USA) and GPR147 (H-001-58, 1:500, Phoenix Pharmaceuticals, Burlingame, CA, USA) for 12 h at 4 °C. After washing with PBS three times, all sections were treated with HRP-conjugated secondary antibody (bs-0295G-HRP, 1:200, Bioss, Woburn, MA, USA), followed by diaminobenzidine (DAB) staining and hematoxylin staining. All images were obtained under bright-field microscopy (Observer Z1, Zeiss, Oberkochen, Germany).

### 5.3. Isolation and Culture of COCs and CCs

After slaughter, the healthy ovarian tissues were collected and transported to the laboratory as soon as possible, as described in our previous study [54]. Cumulus–oocyte complexes (COCs) were collected from antral follicles (diameter in 3~8 mm) using a 10-mL syringe with a 16-gauge needle, which contained 1 mL of thermogenic PBS (phosphate buffered saline). COCs surrounded with at least three layers of compact CCs and homogeneous cytoplasm were chosen and washed three times with PBS. According to our experimental design, parts of selected COCs were dealt with PBS containing 0.2% (*w*/*v*) hyaluronidase and subjected to mechanical separation by repeated pipetting to collect yak CCs. These CCs were washed in PBS with 1% FBS (fetal bovine serum) after centrifugation at 1000 rpm for 3 min. All these cells were resuspended in DF12 (DMEM/Ham’s F12, Gibco) and added with 10% FBS, 1% streptomycin, and 1% penicillin. Approximately 1.0 × 10^5^ cells were cultured into six-well plates under 5% CO_2_ at 37.5 °C. To screen the optimal RFRP-3 dosage and treatment time, CCs were incubated with different concentrations of RFRP-3 (0, 10^−10^ mol/L, 10^−8^ mol/L, and 10^−6^ mol/L) and times (12 h, 24 h, and 36 h). Additionally, the effects of GRP147 on the actions of RFRP-3 in yak CCs were investigated by co-treating CCs with GRP147 antagonist RF9 (10 μM).

For IVM, parts of COCs were randomly divided into several groups (25~30 /group), individually cultured in droplets with IVM medium (TCM199 added with 1 μg/mL 17β-estradiol, 10% FBS, 0.5 μg/mL FSH and 0.5 μg/mL LH) containing various dosages of RFRP-3 (0, 10^−10^, 10^−8^, and 10^−6^ mol/L) with or without RF9, and covered with mineral oil (Vitrolife, Sweden) at 38.5 ºC under 5% CO_2_ for 24 h.

### 5.4. Immunofluorescence Assay

FSHR was used to identify the CCs as described in our previous study [54], and the expression patterns of GnIH/RFRP-3 and GRP147 were also detected. In brief, after seed and culture for 48 h, the CCs were fixed with 4% paraformaldehyde in PBS (pH = 7.2) and then permeabilized in 0.2% Triton X-100 for 20 min at RT (room temperature). The cells were then incubated with primary antibody for FSHR (bs-20659R, 1:200, Bioss, Woburn, MA, USA), GnIH (N2176-47, 1:500, Biological, Swampscott, MA, USA), or GPR147 (H-001-58, 1:200, Phoenix Pharmaceuticals, Burlingame, CA, USA) for 12 h at 4 °C, followed by incubation with FITC-conjugated secondary antibody (bs-0295G-FITC, 1:500, Bioss, Woburn, MA, USA) for 30 min at RT. Furthermore, CCs that were not incubated with primary antibodies but only incubated with secondary antibodies were used as the negative controls. These nuclei were counterstained by DAPI (5 g/L) for 5 min. All images were captured at the same intensity of fluorescence by using a confocal microscope (LSM800, Zeiss, Oberkochen, Germany).

### 5.5. Proliferation and Apoptosis Analysis

The proliferation of yak CCs was evaluated with the EdU assay as described in our previous study [54]. In brief, cultured yak CCs (2 × 10^3^ cells/well) were washed with PBS three times, and 50 mM EdU (RiboBio, Guangzhou, China) was supplemented and co-incubated for 3 h at a final concentration of 50 mmol/L. Subsequently, these nuclei were stained with 5 g/L DAPI for 5 min. Finally, the cells were counted, and photos were obtained under the same intensity of fluorescence using a fluorescence microscope (Observer Z1, Zeiss, Oberkochen, Germany).

The apoptotic rate of yak CCs was detected with the TUNEL detection kit (Yeasen, Shanghai, China) according to the manufacturer’s instructions. Specifically, after being fixed with 4% paraformaldehyde for 4 h, yak CCs were incubated with DNase I (10 U/mL) for 10 min, and CCs were treated with dATP and TdT for 10 min at RT. These cellular nuclei were stained with 5 g/L DAPI. Finally, photos were obtained under the same intensity of fluorescence using a fluorescence microscope (Observer Z1, Zeiss, Oberkochen, Germany).

### 5.6. Assay of Steroid Hormone Content

The contents of the hormones P_4_ and E_2_ in the supernatant of culture media were determined with an E_2_ ELISA kit (ml058533, Mlbio, Shanghai, China) and P_4_ ELISA (enzyme-linked immunosorbent) assay kit (ml057778, Mlbio, Shanghai, China) according to the manufacturer’s instructions with minor modifications. In brief, the OD (Optical Density) of each sample was read at 450 nm by using a microtiter plate reader within 10 min. The inter-assay and intra-assay variations for E_2_ were 4.7% and 5.9%, and 5.4% and 7.5% for P_4_.

### 5.7. Assessment of Oocyte Meiotic Progression

After IVM and treatment with 0.2% hyaluronidase, denuded oocytes were stained with 5 g/L DAPI in PBS (Invitrogen, Waltham, MA, USA). The nuclear morphology was observed and evaluated to determine the meiotic stages under fluorescence microscopy (Observer Z1, Zeiss, Oberkochen, Germany). At 6 h of IVM, oocytes were classified as the GV (germinal vesicle) or GVBD (germinal vesicle breakdown) stage before or after germinal vesicle breakdown according to the nuclear status, respectively. After 12 or 24 h of IVM, oocytes were classified as the MI (metaphase I) or MII (metaphase II) stage according to the nuclear morphology.

### 5.8. Parthenogenetic Activation

After 24 h of IVM, oocytes were activated through parthenogenesis, as described in our previous study [54]. In brief, denuded oocytes belonging to the MII stage (with the first polar body) were chosen for parthenogenesis from each group. After washing twice with PBS, the selected MII oocytes were loaded into 5 μM ionomycin for 5 min and then incubated for 4 h in mSOF (modified synthetic oviductal fluid) supplemented with 2 mM dimethylaminopurine. Subsequently, the activated potential parthenotes were cultured in mSOF droplets (20~30 oocytes a group), which were covered with mineral oil (VitroLife, Kungsbacka, Sweden) at 38.5 °C under 5% CO_2_ for 7 days. The 2-cell (cleavage), 8-cell, and blastocyst developmental rates were observed at 24, 72, and 168 h of in vitro incubation.

### 5.9. RT-qPCR

The total RNA of CCs was extracted with the Trizol reagent (Invitrogen, Carlsbad, CA, USA) according to the manufacturer’s instructions. Complementary DNA (cDNA) transcription and RT-qPCR were performed as previously described [54]. RT-qPCR was performed on a CFX96 Real-Time PCR (BioRad, Hercules, CA, USA) in a 20 μL reaction volume using SYBR Green PCR Master Mix (Invitrogen, Carlsbad, CA, USA) and 2.5 pmol primers. The RT-qPCR procedure comprised 95 °C for 10 min, 95 °C for 15 s, and the annealing temperature for 1 min for 40 cycles. The sequences of primers used for RT-qPCR were designed by Primer 5.0 software, and they are presented in Appendix A. The specificity of each PCR amplification was verified by melting curve analysis. The mRNA relative expression levels of target genes were calculated using the 2^–ΔΔCt^ method, and the housekeeping gene *Gapdh* (glyceraldehyde-3-phosphate dehydrogenase) was used to normalize the expression level of each target gene. The average expression level of each gene from the control group was set to 1 for easy comparison. The experiment was repeated less than three times.

### 5.10. Statistical Analysis

All quantitative data, proliferation rate, apoptosis rate, oocyte maturation, and embryonic developmental data were expressed as the means ± S.E.M (standard error of the mean), where differences were assessed by one-way ANOVA followed by the Bonferroni multiple-comparison test with SPSS 19.0. If without a special mark, each experiment was repeated three times. Statistical significance was set at *p* < 0.05, and all statistical analyses were performed using GraphPad Prism version 5.0.

## Figures and Tables

**Figure 1 ijms-24-07000-f001:**
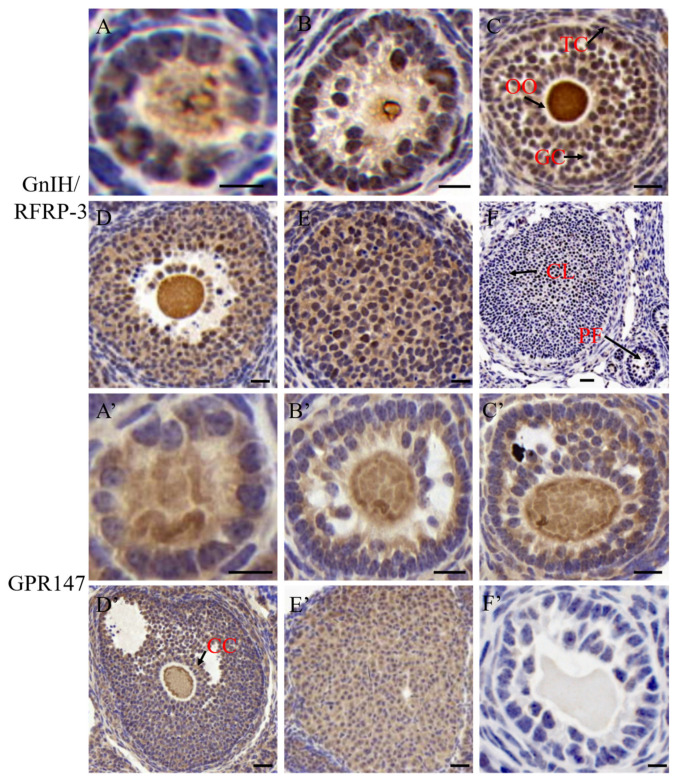
The expression profiles of GnIH/RFRP-3 and GPR147 in growing follicles and corpus luteum. Immunohistochemical (IHC) analysis showed the expression patterns of GnIH/RFRP-3 and GPR147 in the primordial follicle (**A**,**A’**), the primary follicle (**B**,**B’**), the preantral follicle (**C**,**C’**), the antral follicle (**D**,**D’**), the corpus luteum (**E**,**E’**), and the negative control (**F**,**F’**). Sections of yak ovary were incubated with GnIH and GPR147 antibodies and then stained with hematoxylin. The scale bar is 50 μm. GC: granulosa cell, TC: theca cell, OO: oocyte, CC: cumulus cell, CL: corpus luteum, PF: preantral follicle.

**Figure 2 ijms-24-07000-f002:**
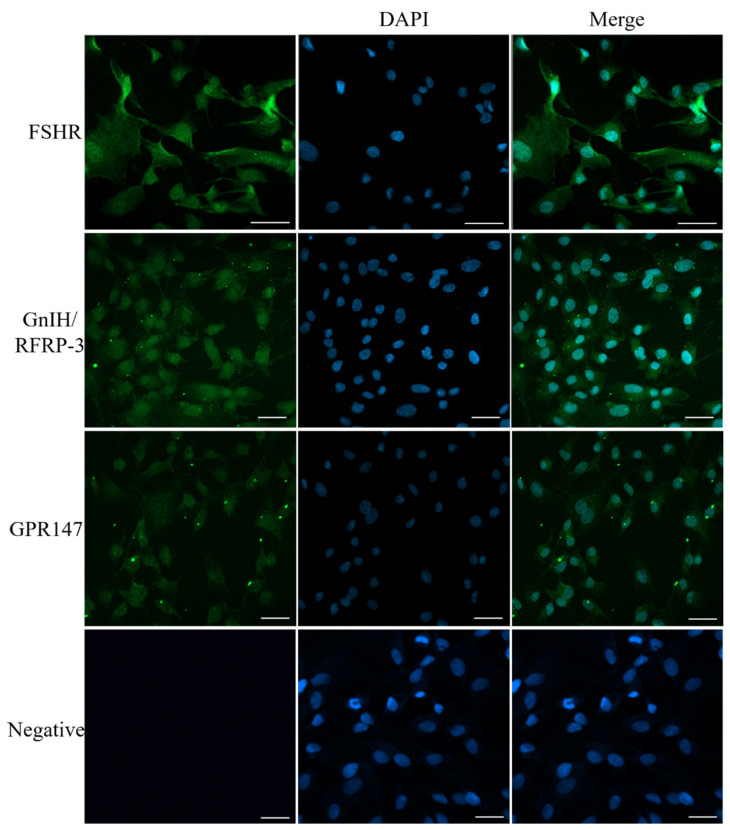
Subcellular localization and expression patterns of GnIH/RFRP-3 and GPR147 in isolated CCs of yak. The expression of FSHR, GnIH/RFRP-3, and GPR147 in CCs were detected by Immunofluorescence. The nuclei of CCs were counterstained with DAPI and observed with a fluorescence microscope. The scale bar is 20 μm (white).

**Figure 3 ijms-24-07000-f003:**
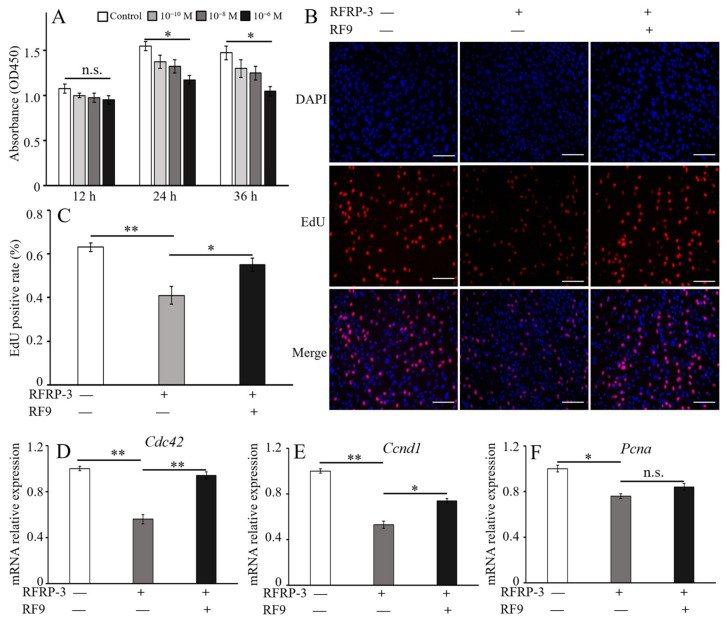
Effects of RFRP-3 and RF9 on proliferation and proliferation-related factors in yak CCs. (**A**) Various dosages (0, 10^−10^, 10^−8^, and 10^−6^ mol/L) of RFRP-3 that influenced the cellular viability of yak CCs were detected with CCK-8 assay. (**B**) The proliferation activity of CCs was evaluated with EdU analysis after combining treatment with RFRP-3 and RF9. The scale (white) is 20 μm. (**C**) The statistical analysis of proliferation rates in cumulus cells. (**D**–**F**) Relative mRNA expression abundances *Cdc42*, *Ccnd* and *Pcna* after yak CCs incubated with 10^−6^ mol/L RFRP-3 and/or 10 μM RF9 for 24 h. Error bars represent mean ± SEM. * indicates a remarkable difference (*p* < 0.05), and ** indicates a considerable difference (*p* < 0.01) from that of the control counterparts. n.s. represents no significant difference.

**Figure 4 ijms-24-07000-f004:**
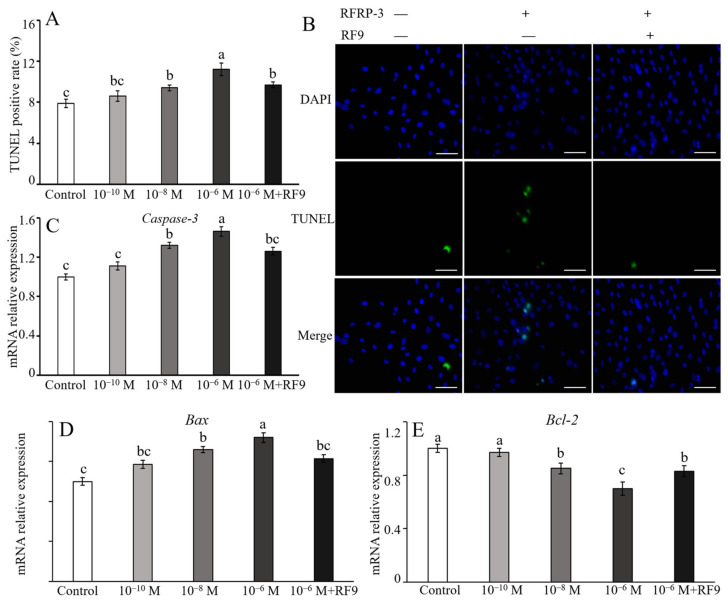
Effects of RFRP-3 and RF9 on apoptosis of yak CCs. (**A**) The statistical analysis of apoptotic rates after CCs were incubated with different doses of RFRP-3 with or without 10 μM RF9. (**B**) Apoptosis of CCs was assessed with TUNEL analysis, and the apoptotic cell of yak CCs were marked in green, and nuclei stained with DAPI (blue). The scale bar (white) is 20 μm. Relative mRNA expression abundances of *Caspase-3* (**C**), *Bax* (**D**), and *Bcl-2* (**E**) were detected by RT-qPCR. Error bars represent mean ± SEM. Different letters (a–c) suggested significantly different (*p* < 0.05) compared to the controls.

**Figure 5 ijms-24-07000-f005:**
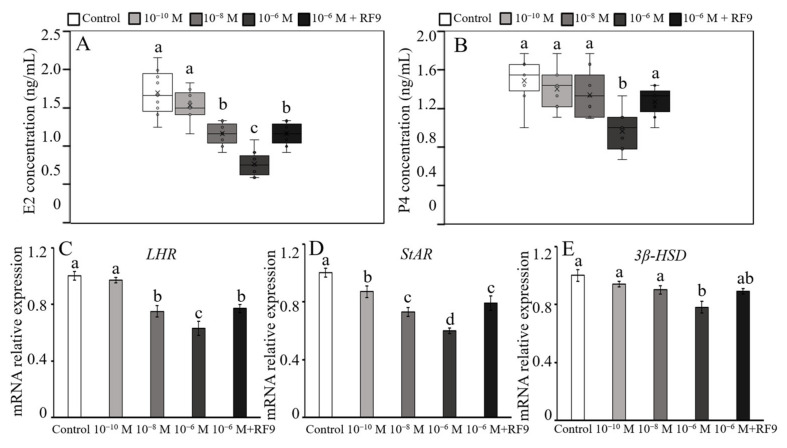
Effects of RFRP-3 and RF9 on the secretion of P_4_ and E_2_ in yak CCs. The concentrations of E_2_ (**A**) and P_4_ (**B**) in the supernatant of yak CCs after treatment with RFRP-3 and/or RF9. (**C**–**E**) Relative expression levels of the steroidogenic pathway genes. Error bars represent mean ± SEM. Different letters (a–d) indicate a remarkable difference (*p* < 0.05) from that of the controls.

**Figure 6 ijms-24-07000-f006:**
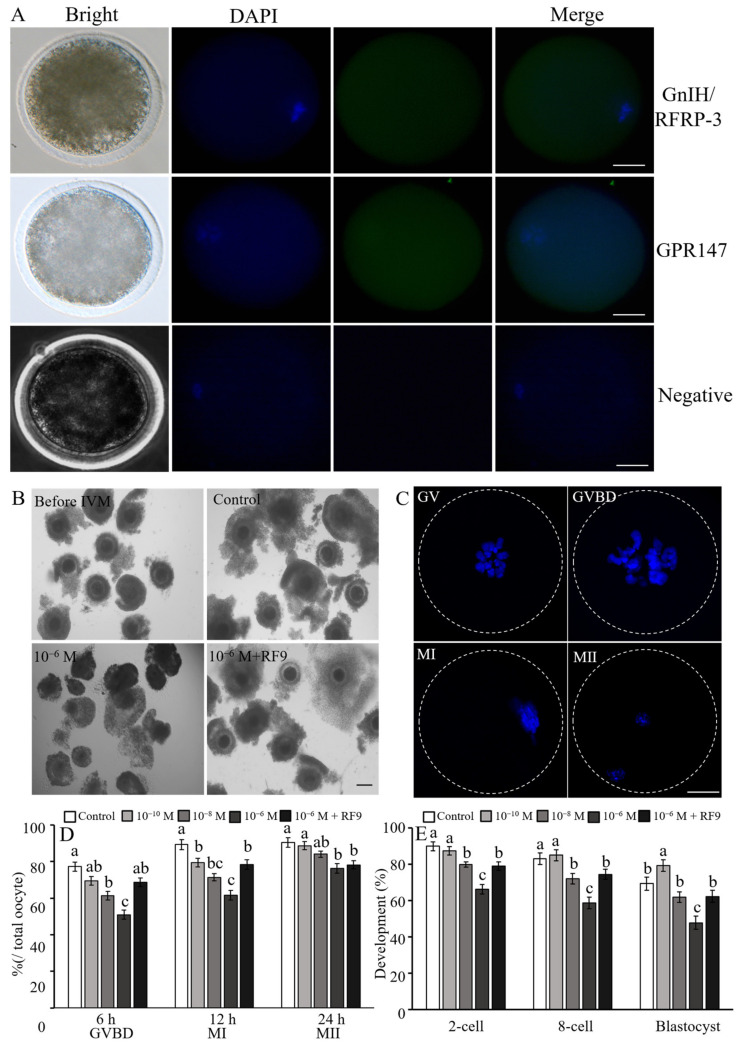
Dose-related effects of RFRP-3 and RF9 on oocyte meiosis progression and subsequent developmental competence. (**A**) Subcellular localization and expression of GnIH and GPR147 in yak oocytes. (**B**) Treatment of RFRP-3 and RF9 impairs the CCs expansion of yak COCs after 24 h IVM. (**C**) The progression of oocyte maturation was evaluated with the nuclear morphology using DAPI staining. (**D**) Statistical analysis of the oocyte meiosis progression after RFRP-3 and/or RF9 treatment. IVM medium was added with different doses of RFRP-3 for 6, 12, or 24 h. GVBD, germinal vesicle breakdown; MI, metaphase I; MII, metaphase II. (**E**) Statistical analysis of the PA embryonic development after oocytes treated with RFRP-3 and RF9. Results are presented as means ± SEM. The scale bar is 50 μm. Different letters (a–c) indicate a significant difference (*p* < 0.05) from that of the controls.

## Data Availability

The data presented in this study are available on request from the corresponding author.

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
