# Peer review of "RFRP-3 Influences Apoptosis and Steroidogenesis of Yak Cumulus Cells and Compromises Oocyte Meiotic Maturation and Subsequent Developmental Competence"

_ijms, 2023, doi:10.3390/ijms24087000_

Round 1

Reviewer 1 Report

Letter to the Authors

Dear Authors, I have read and reviewed your manuscript. I found the topic extremely interesting and worth studying. However, the way of presentation of your study is not satisfactory and has to be improved, before potential publication. Below I have listed all my doubts and concerns.

Major concerns:

1.      The manuscript requires essential English language grammar and style correction. Some sentences are difficult to understand and of a very bad style. I would like the Authors to ask a professional translator or a native speaker for help with correcting the manuscript.

2.      Lines 119-121: The citation [26] is improper. The Authors wrote “in pigs” but the cited paper is about mice.

3.      Fig. 1: Why the Authors applied the control only for corpus luteum. I would prefer to see the cross-section of the whole ovary with CL and follicles.

4.      Fig 2A and 6A: Why the Authors showed no negative controls? Similarly to IHC, there should be proper negative controls applied and presented in this figure.

5.      Fig 2B and C: What was the reason for comparing the abundance of GnIH/RFRP-3 and GPR147? It provides no important information. For more, from a technical point of view, the comparison of two different proteins (different antibodies) and transcripts (different primers and reaction efficiency) is an incorrect approach. It would be much more interesting to compare the same transcripts/proteins between different physiological statuses (i.e. phases of the oestrous cycle/ pregnancy). Furthermore, F-IHC is not a quantitative method only a qualitative one, even though the Authors applied the same parameters during picture taking.

6.      Line 284: The shortcuts “GV, GVBD, MI and MII” have to be described when first time used.

7.      Line 371: Why the Authors wrote “protein levels”? They did not study the level of protein, only mRNA.

8.      Lines 380-384: The Authors mention rats, pigs and cattle, but the citation only refers to pigs. What about checking all citations and correcting them?

9.      In the “Discussion” section the Authors use “low dosage” and “high dosage” referring to other papers, but I could not find any comparison to the dosages used in this study. Are cited “high doses” comparable to those used in the study?

10.   Line 410: What do you mean by “improved E2 concentration”? In this case, “increased” could be more acceptable if only the sentence was correct in grammar and style.

11.   Line 421 and other parts of the manuscript: LHR is not a part of the steroidogenesis pathway! It is not directly involved in steroid production.

12.   Several parts of the manuscript are lacking proper citations. Some citations are improper.

13.   “Material and Methods” unit:

-        Provide information on the exact number of animals used in the study, the number of biological and technical repetitions in each experiment

-        The Authors provided no information on animals: number, age, phase of the cycle, health state, farming conditions, etc. This information seems to be important for the appropriate understanding of the results.

-        In my opinion, the order of sections in this unit should correspond to the order of results presented earlier.

-        Please, confirm if the number of cells was 1×105 and not 1×106.

-        Section 5.1: provide the number of Ethical Committee approval.

-        Is the sequence of human RFRP-3 the same as yak’s? Have the Authors compare these sequences i.e with BLAST.

-        What was the basis for chosen RFRP-3 doses? Have you checked the level of RFRP-3 in the follicular fluid? Are the chosen doses similar to the physiological concentrations of the peptide?

-        Line 530: What do you mean by “co-incubated”? Was it double/triple staining? It seems that there were separate stainings for each antigen, so it is misleading.

-        Line 531: there should be “or” instead of “and” as you did not perform double staining.

-        Sections 5.3 and 5.4: How were negative controls prepared? There should be proper information in the method description. Why for F-IHC you showed no negative controls? Please add both, the information in the text and the appropriate images.

-        Line 540: “three times” and “co-incubated”.

-        Line 552: Description for P4 and E2 should be placed when shortcuts were first time used. The same for all shortcuts in the manuscript.

-        Lines 557-558: What exactly were the intra- and inter-assay variations?

-        Unit 5.9: the provided information on RT-PCR is insufficient. Please, rewrite it according to the MIQE guidelines for RT-PCR presentation. In the case of newly designed primers, please, provide data for primer validation, at least the Slope (S) and reaction efficiency. For more, according to the newest requirements, there should be at least two reference genes. Please provide data on GAPDH expression stability in the compared groups. For more, you provided no information on reaction conditions

-        Unit 5.10: Which statistical tests exactly you used in each experiment, please specify

Minor corrections:

1.      Line 20: delete “in the present study”;

2.      Across the manuscript, please write “in vitro/in vivo” in italics;

3.      Lines 59-65: This sentence has to be rewritten as it makes no sense;

4.      Line 153: delete “expression”;

5.      Line 160: delete “and… fluorescence”;

6.      Decide if you use “co-treated” or “cotreated”;

7.      Decide if you use “P4/E2” or P4/E2”;

8.      Line 307: “activated”;

9.      Line 319: “the suppression of”;

10.   Line 350: Please avoid using phrases like “decision of follicles” as follicles are not making any decisions, they are not humans;

11.   Line 503: “DMEM” not “DEME” 

Reviewer 2 Report

In the current study the authors assessed the biological functions of exogenous RFRP-3 on the apoptosis and steroidogenesis of yak cumulus cells (CCs) and the developmental potential of the derived yak oocytes. They observed that RFRP-3 dose-dependently elevated the expression of apoptosis markers (Caspase and Bax), whereas the expression levels of steroidogenesis related factors (LHR, StAR, 3β-HSD) exhibited a downregulation in dose-dependent manner. I have some comments before considering for further publication.

Major comments

1. Pleaase clarify how the treatment protocol was established: concentration and timings declared in line 195. Did the authors perform a dose-responese curve on preliminary data?

2. The number of approval from Ethic Committee should be provided

Minor comments (english revision)

1. Line 51: necessary

2. Line 58: compared

3. Lines 499, 540 and 87: three times

4. Line 517: detected

5. Lines 565, 566 according to

Author Response

In the current study the authors assessed the biological functions of exogenous RFRP-3 on the apoptosis and steroidogenesis of yak cumulus cells (CCs) and the developmental potential of the derived yak oocytes. They observed that RFRP-3 dose-dependently elevated the expression of apoptosis markers (Caspase and Bax), whereas the expression levels of steroidogenesis related factors (LHR, StAR, 3β-HSD) exhibited a downregulation in dose-dependent manner. I have some comments before considering for further publication.

Answer: Thanks for your evaluation and suggestion.

Major comments

  1. Please clarify how the treatment protocol was established: concentration and timings declared in line 195. Did the authors perform a dose-responese curve on preliminary data?

Answer: Thanks for your evaluation and suggestion. In this study, various dosages of RFRP-3 (0, 10-10, 10-8, and 10-6 mol/L) with or without RF9 were used to treat CCs. Before our form experiment, we referred to relevant experimental designs on other species (such as pig, bovine) to establish the experimental protocol. Furthermore, we also did a preliminary experiment to preliminary evaluation of the effects of different concentrations and treatment times on CCs. After that, the viability of CCs was detected by the EdU, and found that the absorbance of the treatment (10-6 mol/L RFRP-3 for 24 h) was significantly higher than others. Thus,we chose this condition (10-6 mol/L RFRP-3 for 24 h) for further analysis. Of course, there may be still not rigorous enough and we will optimize our design, such as perform a dose-responese curve.

  1. The number of approval from Ethic Committee should be provided

 Answer: Thanks for your suggestion. We have added as “~~~and were approved by the Southwest Minzu University Ethical Committee (approval number: 2020A017)”. Actually, we have stated that in the part of “Institutional Review Board Statement”.

Minor comments (english revision)

  1. Line 51: necessary

Answer: Thanks for your suggestion. We have revised.

  1. Line 58: compared

Answer: Thanks for your suggestion. We have revised.

  1. Lines 499, 540 and 87: three times

Answer: Thanks for your suggestion. We have revised as “three times”

  1. Line 517: detected

Answer: Thanks for your suggestion. We have revised.

  1. Lines 565, 566 according to

Answer: Thanks for your suggestion. We have revised.

Round 2

Reviewer 1 Report

I am not absolutely satisfied with the authors' answers. Some of my suggestions should be added to the manuscript, others are still lacking for information I asked for. Please, find my concerns placed as comments in the pdf file of the cover letter. They address the marked fragments of the answers strictly.

Author Response

Reviewer 1

I am not absolutely satisfied with the authors' answers. Some of my suggestions should be added to the manuscript, others are still lacking for information I asked for. Please, find my concerns placed as comments in the pdf file of the cover letter. They address the marked fragments of the answers strictly.

Answer: Thanks for your evaluation and suggestion. Some suggestions may not be fully understood that part of our answers do not meet your expectations. I am so sorry for that, and we have answered all your suggestions, especially all comments in the PDF. Thank you again for your suggestion.

Reviewer 2 Report

All comments have been addressed

Author Response

Thank you for your evaluation and approval.

Round 3

Reviewer 1 Report

I am satisfied with the corrections made to the manuscript.